# Clinical progression, disease severity, and mortality among adults hospitalized with COVID-19 caused by the Omicron and Delta SARS-CoV-2 variants: A population-based, matched cohort study

**COVID-19 Omicron Delta study group**[¶]

¶ Membership of the COVID-19 Omicron Delta study group is listed in the Acknowledgments.
* zitta.barrella.harboe@regionh.dk; casper.roed@regionh.dk

## Abstract

### Background

To compare the intrinsic virulence of the severe acute respiratory syndrome coronavirus 2 (SARS-CoV-2) omicron variant with the delta variant in hospitalized adults with coronavirus disease 2019 (COVID-19).

### Methods

All adults hospitalized in the Capital Region of Copenhagen with a positive reverse transcription polymerase chain reaction test for SARS-CoV-2 and an available variant determination from 1 September 2021 to 11 February 2022. Data from health registries and patient files were used. Omicron and Delta patients were matched (1:1) by age, sex, comorbidities, and vaccination status. We calculated crude and adjusted hazard ratios (aHRs) for severe hypoxemia and mortality at 30 and 60 days.

### Results

1,043 patients were included. Patients with Omicron were older, had more comorbidities, were frailer, and more often had three vaccine doses than those with Delta. Fewer patients with Omicron developed severe hypoxemia than those with Delta (aHR, 0.55; 95% confidence interval, 0.38–0.78). Omicron patients exhibited decreased aHR for 30-day mortality compared to Delta (aHR, 0.61; 0.39–0.95). Omicron patients who had received three vaccine doses had lower mortality compared to Delta patients who received three doses (aHR, 0.31;0.16–0.59), but not among those who received two or 0–1 doses (aHR, 0.86; 0.41–1.84 and 0.94; 0.49–1.81 respectively). Similar findings were observed for mortality at 60 days. Similar outcomes were obtained in the analyses of 316 individually matched patients.

**Data Availability Statement:** Data collected for this study, including individual participant data and a data dictionary defining each field in the set, are not publicly available because they contain potentially identifying and sensitive patient information. Upon request from investigators, data can be made available. Such requests should include a study protocol with clear hypotheses as well as the required permissions from the appropriate local authorities according to Danish law on Health, § 46 stk. 2, and be sent to the principal investigator, which will then be reviewed by the steering committee. If the hypothesis complies with the authorizations for this study, and is judged to be valid, a data transfer agreement will precede transfer of de-identified data. Alternatively, data requests may be sent to serum@ssi.dk.

**Funding:** The authors received no specific funding for this work.

**Competing interests:** I have read the journal's policy and the following authors of this manuscript have declared that no competing interests exist: JGH, CKJ, PB, TGK, FIK, ANAA, SLM, BLJ, BBB, CS, TLN, LVH, BLM, AB, ME, PHB, EMJP, GBE, AMD, AS, IHMM, NZJ, PTP, MPGJ, TIP, RE, HESSR, MB, US, HA, MRS, PS, SA, JUSJ, KB, KMJ, MJSK, TL, UVS, MGA, SR, NID, PR, NR, DP, AK, SJ, LEK, CL, BBH, OK, SRO, STS, AP, NK, MSP, LS, MV, LEC, MS AC, JF, AF, JW, RL, MR, TL, CR and TKF. I have read the journal's policy and the following authors of this manuscript have following competing interests: ZBH has received research grants from Independent Research Fund Denmark (grant nr. 0134-00257B) and Lundbeck Foundation (grant nr. R349-2020-835). MvW was partially supported by the Independent Research Fund Denmark (grant # 8020-00284), and Carlsberg Foundation, Semper Ardens Research Project (grant # CF20-0046). LN received travel grants from MSD, GSK and Gilead. TB has received grants from Novo Nordisk Foundation, Lundbeck Foundation, Simonsen Foundation, GSK, Pfizer, Gilead, Kai Hansen Foundation and Erik and Susanna Olesen's Charitable Fund and personal fees from GSK, Pfizer, Boehringer Ingelheim, Gilead, MSD, Pentabase ApS, Becton Dickinson, Janssen and Astra Zeneca, all outside the submitted work.

## Conclusions

Among adults hospitalized with COVID-19, those with Omicron had less severe hypoxemia and nearly 40% higher 30- and 60-day survival, as compared with those with Delta, mainly driven by a larger proportion of Omicron patients vaccinated with three doses of an mRNA vaccine.

## Introduction

On 26 November 2021, the World Health Organization designated the severe acute respiratory syndrome coronavirus 2 (SARS-CoV-2) B.1.1.529 variant, Omicron, as the fifth variant of concern [1]. Omicron was rapidly emerging as the dominant SARS-CoV-2 variant circulating worldwide [2]. Early epidemiological studies from South Africa [3, 4], the United States [5, 6], and Europe [7–9] indicated that the Omicron variant was associated with a significantly reduced risk of hospitalization, intensive care unit (ICU) admission, and death compared with the Delta variant. However, these results originated from populations with high levels of immunity, either due to high vaccination coverage or infection rates. Therefore, the conclusions regarding the intrinsic virulence and severity of coronavirus disease 2019 (COVID-19) due to the SARS-CoV-2 Omicron variant were considered uncertain.

Although the absolute risks of ICU admission and death associated with Omicron infection were reportedly lower than those associated with Delta infection, global data indicated that deaths temporarily increased after Omicron became the predominant variant worldwide [2, 10]. Some of this excess mortality could be explained by larger proportions of unvaccinated older adults in the populations [8, 10]. Furthermore, increasing evidence suggests that a third dose of an mRNA vaccine, following either homologous or heterologous vaccination, leads to higher levels of neutralizing antibodies and increased clinical protection against hospitalization and death caused by either variant [11–14]. However, few clinical studies have described the severity of the disease caused by the Omicron variant in patients hospitalized due to COVID-19, and these studies have produced conflicting results [3, 4, 8].

We hypothesized that in hospitalized patients, the Omicron variant is associated with a milder course of COVID-19, compared with the Delta variant. To better understand the differences in clinical progression and severity of COVID-19 associated with these variants, we conducted a large retrospective cohort study by reviewing medical charts of patients hospitalized with SARS-CoV-2 infection, investigating factors linked to poor clinical outcomes for each variant. Furthermore, we individually matched patients within the cohort by age, sex, underlying comorbidities, and vaccination status, allowing us to explore the intrinsic differences between the two variants following hospitalization.

## Methods

### Study design and study population

A retrospective, multicenter, matched cohort study in the Capital Region of Copenhagen, Denmark, with a population of approximately 1.8 million. As of 1 March 2022, >1 million SARS-CoV-2 infections were confirmed in the region since the first case was reported on 26 February 2020. The first Omicron case was detected on 27 November 2021, with a peak incidence of approximately 1,000 daily cases per 100,000 population in mid-January 2022 [15]. The design and reporting of this study follows the Strengthening the Reporting of Observational Studies in Epidemiology (STROBE) statement [16].

## Setting

Patients admitted to any of the nine university hospitals in the region with a positive SARS-CoV-2 reverse transcription polymerase chain reaction (RT-PCR) test from 1 September 2021 to 11 February 2022 were eligible for inclusion. We chose this period to ensure the inclusion of patients admitted when the Delta variant (September to November) and Omicron variant (December to February) were predominant. The prevalence of the Omicron BA.1 sublineage peaked at 71.9% of all sequenced samples in late December, and as of 1 February the Omicron BA.2 sublineage accounted for 89.2% of all sequenced samples [17]. There were no significant changes in the national treatment guidelines for COVID-19 during the study period [18], and the health care system was not considered overwhelmed. By 22 May 2022, the national vaccination coverage was 90% for a first dose, 89% for a second dose, and 76% for a third dose [19]. More than 90% of individuals received an mRNA vaccine [17] (Comirnaty, BNT162b2 mRNA; BioNTech [Mainz, Germany]-Pfizer [New York, NY, USA] or Spikevax, mRNA-1273; Moderna, Cambridge, MA, USA).

## Data sources and data collection

Data from the National Patient Register [20], the Danish Civil Register [21], the Danish Vaccination Register [22], and the national COVID-19 surveillance system at Statens Serum Institut (Copenhagen, Denmark) [9, 17] were used to identify all individuals aged ≥18 years who were hospitalized for >12 h within 14 days of a positive RT-PCR test for SARS-CoV-2 and from whom variant information was available. Surveillance and screening algorithms of SARS-CoV-2 variants in Denmark have been described elsewhere [9, 17]. Variant determination was based on whole-genome sequencing or variant-specific RT-PCR performed at Statens Serum Institut or locally at the departments of clinical microbiology in the region.

Medical doctors with experience in caring for COVID-19 patients reviewed all medical records using a predefined electronic data capture instrument (REDCap) [23]. All clinical data that were not available from national registries were collected manually. These included characteristics at admission, laboratory results, and treatments. The patients' hospitalization courses were categorized as mild, moderate, severe, or critical COVID-19, or attributable to causes other than COVID-19 (with SARS-CoV-2 infection being incidental, i.e., asymptomatic SARS-CoV-2 infection) [18, 24]. We excluded individuals aged <18 years, those with a prior positive RT-PCR SARS-CoV-2 test result, those with a positive RT-PCR SARS-CoV-2 test result >48 h after admission (presumed nosocomial infections), those who were hospitalized in other regions, those transferred from outside to a hospital in the Capital Region, and those from whom vaccination records were unavailable.

## Participants

After the medical chart review, patients fulfilling the inclusion criteria who were considered hospitalized due to COVID-19 were included in our study population. We also constructed a second study population by matching all Omicron patients with Delta patients in a 1:1 ratio, with exact matching by age, sex, number of comorbidities that were considered risk factors for severe COVID-19 (0, 1, 2, > 3, as described below), and vaccination status (0, 1, 2, or 3 doses, regardless of the particular vaccines administered).

## Variables, outcomes, and exposures

*Characteristics at admission*: age (years), sex, pre-admission frailty level (defined as 1, no need for help; 2, limited need for assistance daily; 3, extensive need for assistance or living in a

nursing home), comorbidities considered to be risk factors for severe COVID-19 (defined as diabetes, body mass index >30, chronic heart disease including hypertension, chronic pulmonary disease, chronic kidney disease, chronic liver disease, active cancer, immunosuppression, cerebrovascular disease, and pregnancy) [18, 25]. Immunosuppression was defined as taking >20 mg prednisolone/daily or equivalent, chemotherapy, biologicals, or methotrexate. Vaccination status was defined as 0, 1, 2, or 3 doses at least 14 days before hospital admission. We calculated the Charlson Comorbidity Index (CCI, Quan) at hospital admission based on hospital discharge diagnoses over the previous 5 years, using the National Patient Register. The CCI was categorized as low (0), moderate (1–2), or high (>2).

*Treatment during hospitalization*: oxygen supplementation therapy (classified as peak after admission: none, <10 L/min, ≥10 L/min), high-flow nasal cannula (HFNC) (FiO2 > 40%, flow > 30 L/min), noninvasive ventilation (NIV), continuous positive airway pressure (CPAP), invasive mechanical ventilation (MV); use of dexamethasone (standard dose, 6 mg; high dose, 8–12 mg), use of monoclonal antibodies before or during admission (e.g., sotrovimab, casirivimab/imdevimab), use of interleukin-6 receptor inhibitor (tocilizumab), use of antivirals (remdesivir, molnupiravir).

*Laboratory results*: biochemical (C-reactive protein, D-dimer, ferritin, and procalcitonin), microbiological (blood and respiratory cultures, and PCR results for atypical and viral pathogens), and radiological (chest X-rays and computed tomography [CT] results).

*Outcomes*: The primary outcomes of interest were: 1) a composite outcome for severe hypoxemia, consisting of the need for ≥10 L/min of supplementary oxygen or HFNC, NIV, CPAP, MV, or ICU admission; 2) death, in hospital or after discharge within 30 or 60 days of COVID-19-related hospital admission. Dates of admission, discharge, and death were verified using national registries.

*Exposures*: Laboratory confirmed positive RT-PCR test results for Omicron or Delta SARS-CoV-2 variants.

## Bias and missing data

We identified and addressed potential sources of bias, please refer to Supplementary Appendix 1 in S1 File and for missing data, please refer to Appendix 2 in S1 File.

## Statistical analyses

Descriptive statistics and distributions of covariates among groups of patients categorized by disease severity and stratified by SARS-CoV-2 variants were calculated. Differences among categorical data were evaluated using $\chi^2$ tests and Fisher's exact test. Student's *t*-test was used for numerical variables, and the Wilcoxon test was used when appropriate.

Patients were followed up for 60 days after the date of COVID-19 hospitalization or until death, whichever occurred first. Follow-up for the composite outcome of severe hypoxemia was restricted to the period of hospitalization. We performed Cox proportional hazard regression analyses after confirming the assumptions of proportional hazards, by checking zero-slope of time-dependent coefficients. We performed univariate and multivariable regression analyses, adjusting for age, sex, pre-admission frailty, CCI, and vaccination status, and reported crude and adjusted hazard ratios (aHRs). Because the vaccination status of patients was likely to differ between patients infected during the Delta and Omicron periods, and because vaccines may differ in efficacy against these variants, interaction between variants and vaccination status was included in a separate analysis. COVID-19 treatment interventions were documented from medical records and were likely to have similar effects on both Omicron and Delta patients. An exception were the monoclonal antibodies where data emerged

showing reduced neutralizing activity of casirivimab/imdevimab against the BA.1 sublineage. Instead, sotrovimab was used but only until the emergence of the BA.2 sublineage where this drug was also shown to be ineffective and ceased to be recommended. During the study period four anti-viral drugs against SARS-CoV-2 were available in Denmark; remdesivir, molnupiravir, casirivimab/imdevimab and sotrovimab. Convalescent plasma was not recommended. Tixageivmab/cilgavivmab, bebtelovimab and nirmatrelvir/ritonavir were not yet available.

Additionally, to optimize group comparisons, we performed a subanalysis in which patients admitted with Delta and Omicron were matched in a 1:1 ratio on age, sex, vaccination status, and underlying comorbidities for the risk of developing severe or critical COVID-19 (as described above). These analyses were further adjusted for frailty at admission.

Statistical significance was set to 0.05 (two-sided). All analyses were performed using R statistical software (ver. 4.2.0; R Development Core Team, Vienna, Austria) [26].

### Ethics statement

Permission for this study was provided by the Danish Health and Medicines Authority (ID:31-1521-263) and the Danish Data Protection Agency (P-2020-375). This study was done with use of administrative register data and by a retrospective study of medical records. All data were fully anonymized before access and according to Danish law ethics approval and informed consent to have data from the study participants medical records are not needed for such research.

## Results

Among 17,676 hospitalized patients who had a positive SARS-CoV-2 RT-PCR test result during the study period, 7,801 had a SARS-CoV-2 variant determined (4,208 Omicron patients and 3,593 Delta patients). Among these, we identified 1,579 individuals (655 Omicron patients and 924 Delta patients) who were eligible for further clinical review. In total, 536 of these patients (338 Omicron patients [51.6%] and 198 Delta patients [21.4%]) were excluded from the analysis, mainly because they were considered incidental (i.e., asymptomatic). Thus, 1,043 patients were included in the study population. For the matched subgroup, we identified 316 patients (S1 Fig in S1 File).

### Baseline characteristics at hospital admission

Table 1 shows the baseline characteristics of the study population at hospital admission. The median age was 72 years and 52.6% (*n* = 549) were male. Compared with Delta, Omicron patients were older, more often of Danish ethnicity, less frequently overweight, and more often had with a history of smoking. Omicron patients had more comorbidities, but the overall frequencies of specific risk factors for severe COVID-19 were similar for both groups, except for active cancer disease, which was more frequent among Omicron patients. In addition, Omicron patients were more likely to be frail and to need extensive assistance or to live in a nursing home. Fewer Omicron patients were unvaccinated, and more had received three vaccine doses. In addition, the median time from the final vaccination to hospitalization was significantly shorter for Omicron than for Delta patients (Table 1).

Paraclinical characteristics during hospitalization are shown in Table 2. Compared with Delta, Omicron patients had lower peak values of C-reactive protein and ferritin levels but similar levels of D-dimer and procalcitonin. Hospitalized Omicron patients were more likely to have bacterial coinfections in the respiratory tract. Bilateral lung consolidations and ground-glass opacities on CT scans were less commonly observed in Omicron than Delta patients.

**Table 1. Baseline characteristics of patients hospitalized with COVID-19 due to SARS-CoV-2 Omicron and Delta variants, according to disease severity (Copenhagen, 1 September 2021 to 11 February 2022; *n* = 1,043).**

| | | Omicron | | | | Delta | | | | |
| --- | --- | --- | --- | --- | --- | --- | --- | --- | --- | --- |
| | | Mild/ Moderate | Severe | Critical | All | Mild/ Moderate | Severe | Critical | All | *p*-value |
| **Total** | | 158 (49.8%) | 121 (38.2%) | 38 (12%) | 317 (30.4%) | 261 (36%) | 296 (40.8%) | 169 (23.3%) | 726 (69.6%) | |
| **Age, median (IQR)** | | 64 (38–80.8) | 78 (71–85) | 74 (66–78) | 74 (57–83) | 68 (47–78) | 72 (53.8–81) | 70 (57–79) | 70 (52–80) | <0.001 |
| **Sex** | Male | 69 (43.7%) | 61 (50.4%) | 25 (65.8%) | 155 (48.9%) | 127 (48.7%) | 157 (53%) | 110 (65.1%) | 394 (54.3%) | 0.12 |
| | Female | 89 (56.3%) | 60 (49.6%) | 13 (34.2%) | 162 (51.1%) | 134 (51.3%) | 139 (47%) | 59 (34.9%) | 332 (45.7%) | |
| **BMI, median (IQR)** | | 24.4 (21.6–28.4) | 24 (21.2–27.9) | 25.6 (22–31.7) | 24.4 (21.5–28.2) | 26.4 (23.1–30.1) | 26.4 (23–31.6) | 27.7 (24–31.2) | 26.8 (23.2–31) | <0.001 |
| **Ethnicity** | Other | 39 (24.7%) | 15 (12.4%) | 9 (23.7%) | 63 (19.9%) | 74 (28.4%) | 83 (28%) | 53 (31.4%) | 210 (28.9%) | 0.003 |
| | Danish | 119 (75.3%) | 106 (87.6%) | 29 (76.3%) | 254 (80.1%) | 187 (71.6%) | 213 (72%) | 116 (68.6%) | 516 (71.1%) | |
| **Smoking status** | Never | 73 (46.2%) | 35 (28.9%) | 10 (26.3%) | 118 (37.2%) | 119 (45.6%) | 125 (42.2%) | 69 (40.8%) | 313 (43.1%) | 0.05 |
| | Current | 11 (7%) | 14 (11.6%) | 4 (10.5%) | 29 (9.1%) | 19 (7.3%) | 31 (10.5%) | 18 (10.7%) | 68 (9.4%) | |
| | Previous | 49 (31%) | 61 (50.4%) | 23 (60.5%) | 133 (42%) | 81 (31%) | 102 (34.5%) | 60 (35.5%) | 243 (33.5%) | |
| | Missing | 25 (15.8%) | 11 (9.1%) | 1 (2.6%) | 37 (11.7%) | 42 (16.1%) | 38 (12.8%) | 22 (13%) | 102 (14%) | |
| **Frailty** | No need for help | 100 (63.3%) | 44 (36.4%) | 14 (36.8%) | 158 (49.8%) | 206 (78.9%) | 201 (67.9%) | 124 (73.4%) | 531 (73.1%) | <0.001 |
| | Limited need for help on a daily basis | 40 (25.3%) | 35 (28.9%) | 9 (23.7%) | 84 (26.5%) | 39 (14.9%) | 64 (21.6%) | 32 (18.9%) | 135 (18.6%) | |
| | Full need for help on a daily basis /lives in a nursing home | 17 (10.8%) | 42 (34.7%) | 15 (39.5%) | 74 (23.3%) | 12 (4.6%) | 28 (9.5%) | 11 (6.5%) | 51 (7%) | |
| **COVID-19 vaccination status*** | None | 31 (19.6%) | 14 (11.6%) | 13 (34.2%) | 58 (18.3%) | 96 (36.8%) | 126 (42.6%) | 86 (50.9%) | 308 (42.4%) | <0.001 |
| | One dose | 3 (1.9%) | 4 (3.3%) | 2 (5.3%) | 9 (2.8%) | 4 (1.5%) | 3 (1%) | 3 (1.8%) | 10 (1.4%) | |
| | Two doses | 42 (26.6%) | 28 (23.1%) | 5 (13.2%) | 75 (23.7%) | 138 (52.9%) | 148 (50%) | 66 (39.1%) | 352 (48.5%) | |
| | Three doses | 82 (51.9%) | 75 (62%) | 18 (47.4%) | 175 (55.2%) | 23 (8.8%) | 19 (6.4%) | 14 (8.3%) | 56 (7.7%) | |
| **Time from last vaccination to admission in weeks, median (IQR)** | Weeks | 15 (8.8–21.7) | 15.6 (10.8–20.6) | 17.4 (13.7–20.6) | 15.6 (9.8–21) | 27 (21.3–30.9) | 29.2 (24.1–33.4) | 28 (22.4–31.5) | 28.2 (22.4–32.2) | <0.001 |
| **Charlson comorbidity index, median (IQR)** | | 1 (0–2) | 2 (0.8–3) | 2 (1–3) | 1 (0–2.5) | 1 (0–2) | 1 (0–2) | 1 (0–2) | 1 (0–2) | <0.001 |
| **Charlson comorbidity index, categorical** | 0 | 62 (39.2%) | 31 (25.6%) | 7 (18.4%) | 100 (31.5%) | 124 (47.5%) | 132 (44.6%) | 77 (45.6%) | 333 (45.9%) | <0.001 |
| | 1–2 | 68 (43%) | 52 (43%) | 20 (52.6%) | 140 (44.2%) | 96 (36.8%) | 111 (37.5%) | 66 (39.1%) | 273 (37.6%) | |
| | 3+ | 28 (17.7%) | 38 (31.4%) | 11 (28.9%) | 77 (24.3%) | 41 (15.7%) | 53 (17.9%) | 26 (15.4%) | 120 (16.5%) | |
| **Severe risk factors for COVID-19** | Pregnancy | 12 (7.6%) | | | 12 (3.8%) | 11 (4.2%) | 2 (0.7%) | 2 (1.2%) | 15 (2.1%) | 0.14 |
| | Diabetes mellitus | 20 (12.7%) | 28 (23.1%) | 16 (42.1%) | 64 (20.2%) | 47 (18%) | 69 (23.3%) | 40 (23.7%) | 156 (21.5%) | 0.68 |

*(Continued)*

**Table 1.** (Continued)

| | | Omicron | | | | Delta | | | | |
| | | Mild/ Moderate | Severe | Critical | All | Mild/ Moderate | Severe | Critical | All | *p*-value |
|---|---|---|---|---|---|---|---|---|---|---|
| | *BMI>30* | 21 (13.3%) | 17 (14%) | 11 (28.9%) | 49 (15.5%) | 48 (18.4%) | 72 (24.3%) | 50 (29.6%) | 170 (23.4%) | <0.01 |
| | *Chronic cardiac disease* | 64 (40.5%) | 70 (57.9%) | 18 (47.4%) | 152 (47.9%) | 114 (43.7%) | 132 (44.6%) | 81 (47.9%) | 327 (45%) | 0.42 |
| | *Chronic pulmonary disease* | 38 (24.1%) | 40 (33.1%) | 9 (23.7%) | 87 (27.4%) | 50 (19.2%) | 80 (27%) | 41 (24.3%) | 171 (23.6%) | 0.19 |
| | *Chronic kidney disease* | 12 (7.6%) | 21 (17.4%) | 7 (18.4%) | 40 (12.6%) | 26 (10%) | 21 (7.1%) | 16 (9.5%) | 63 (8.7%) | 0.06 |
| | *Chronic liver disease* | 1 (0.6%) | 1 (0.8%) | 2 (5.3%) | 4 (1.3%) | 6 (2.3%) | 5 (1.7%) | 1 (0.6%) | 12 (1.7%) | 0.79 |
| | *Active cancer* | 21 (13.3%) | 21 (17.4%) | 13 (34.2%) | 55 (17.4%) | 27 (10.3%) | 28 (9.5%) | 16 (9.5%) | 71 (9.8%) | <0.001 |
| | *Immunosuppression*** | 25 (15.8%) | 15 (12.4%) | 6 (15.8%) | 46 (14.5%) | 43 (16.5%) | 26 (8.8%) | 14 (8.3%) | 83 (11.4%) | 0.18 |
| | *Cerebrovascular disease* | 22 (13.9%) | 30 (24.8%) | 9 (23.7%) | 61 (19.2%) | 36 (13.8%) | 52 (17.6%) | 22 (13%) | 110 (15.2%) | 0.12 |
| **Number of severe COVID-19 risk factors, mean (IQR)** | | 1.5 (1–2) | 2 (1–3) | 2.4 (1–3) | 1.8 (1–3) | 1.6 (1–2) | 1.6 (1–3) | 1.7 (1–3) | 1.6 (1–2) | 0.08 |
| **Time from symptoms to admission, median (IQR)** | *Days* | 1 (0–4) | 2 (1–4.5) | 4 (2–7) | 2 (0–4) | 5 (2–8) | 5 (2–8) | 7 (4–9) | 6 (2–8) | <0.001 |

* Vaccination status was defined as 0, 1, 2, and 3 doses at least 14 days before hospital admission.

** Immunosuppression was defined as using prednisolone > 20 mg/daily or equivalent, chemotherapy, biologicals, or methotrexate (MTX).

In-hospital outcomes, treatments, and death within 30 and 60 days of admission are shown in Table 3. All patients had complete follow-up for the outcomes of interest. The median duration between first symptoms and hospitalization was shorter for Omicron (2 days; IQR, 0–4) than for Delta patients (6 days; IQR, 2–8). In addition, nearly half of Omicron patients did not require oxygen during admission, compared with 35% of Delta patients. Furthermore, fewer hospitalized Omicron patients required ≥10 L/min of supplementary oxygen, HFNC, NIV, CPAP, MV, or ICU admission. Fewer Omicron patients received dexamethasone, interleukin-6 receptor blocker, or anticoagulants, but there were no significant differences between the two patient groups with regard to monoclonal antibody or prescribed antibiotic use (Table 3).

### Risks of severe hypoxemia during admission and death at 30 and 60 days

Overall, the Omicron variant was associated with a lower risk of severe hypoxemia (aHR, 0.55; 95% confidence interval, 0.38–0.78) than the Delta variant (Table 4). Crude case mortality at 30 and 60 days was similar between hospitalized Omicron and Delta patients (14.2% vs. 13.6%, *p* = 0.845 at 30 days, and 16.7% vs. 16.3%, *p* = 0.851 at 60 days, respectively) (Table 3).

Among the entire population, in the multivariable analyses the Omicron variant was associated with a lower risk of 30-day mortality than the Delta variant (aHR, 0.61; 95% CI, 0.39–0.95) (Table 4). In patients who received three vaccine doses, Omicron was associated with lower 30-day mortality (aHR, 0.31; 0.16–0.59) in the subanalysis where interaction between variants and vaccination status was included (Table 4). There was no statistically significant difference between the variants for those who received two doses (aHR, 0.86; 0.41–1.84) or 0–1 dose (aHR, 0.94; 0.49–1.81) (Table 4). Higher comorbidity levels, increased frailty, and older age were all associated with an increased 30-day risk of mortality (Table 4). Similar estimates

**Table 2. Laboratory results of patients hospitalized with COVID-19 due to SARS-CoV-2 Omicron and Delta variants, according to disease severity (Copenhagen, 1 September 2021–11 February 2022; *n* = 1,043).**

| | Omicron | | | | Delta | | | | |
| | Mild/ Moderate | Severe | Critical | All | Mild/ Moderate | Severe | Critical | All | *p*-value |
|---|---|---|---|---|---|---|---|---|---|
| **Total** | 158 (49.8%) | 121 (38.2%) | 38 (12%) | 317 (100%) | 261 (36%) | 296 (40.8%) | 169 (23.3%) | 726 (100%) | |
| **Biochemistry** | | | | | | | | | |
| *CRP, median (IQR)** | 30 (8–66) | 91 (47–153) | 196 (139–292) | 62 (22–139) | 41 (18–91) | 93 (51–140) | 158(110–211) | 62 (40–150) | 0.01 |
| *D-dimer, median (IQR)** | 1 (0.4–2.6) | 1.3 (0.8–2.3) | 2.7 (1.7–6.6) | 1.6 (0.8–3) | 0.6 (0.3–1) | 0.9 (0.5–1.6) | 1.7 (0.9–4) | 1.6 (0.6–2.1) | 0.89 |
| *Ferritin, median (IQR)** | 147 (60–341) | 212 (123–485) | 350 (201–807) | 221 (99–529) | 345 (138–563) | 557 (259–1340) | 1205 (587–2000) | 657 (302–1545) | <0.001 |
| *Procalcitonin, median (IQR)** | 0.1 (0.1–0.4) | 0.2 (0.1–1.5) | 0.7 (0.3–7) | 0.3 (0.1–1.1) | 0.2 (0.1–0.4) | 0.1 (0.1–0.4) | 0.5 (0.1–1.7) | 0.3 (0.1–0.8) | 0.35 |
| **Microbiological findings** | | | | | | | | | |
| *Positive blood cultures* | 3 (1.9%) | 15 (12.4%) | 8 (21.1%) | 26 (8.2%) | 5 (1.9%) | 8 (2.7%) | 32 (18.9%) | 45 (6.2%) | 0.29 |
| *Positive respiratory cultures* | 5 (3.2%) | 24 (19.8%) | 21 (55.3%) | 50 (15.8%) | 9 (3.4%) | 20 (6.8%) | 44 (26%) | 73 (10.1%) | 0.01 |
| *Atypical bacterial pneumonia*** | | 2 (1.7%) | | 2 (0.6%) | 3 (1.1%) | 1 (0.3%) | | 4 (0.6%) | 1 |
| **Chest X-ray performed in** | 85 (53.8%) | 113 (93.4%) | 34 (89.5%) | 232 (73.2%) | 178 (68.2%) | 266 (89.9%) | 157 (92.9%) | 601 (82.8%) | <0.001 |
| **Chest X-ray findings** | | | | | | | | | |
| *Normal* | 58 (68.2%) | 33 (29.2%) | 2 (5.9%) | 93 (40.1%) | 80 (44.9%) | 47 (17.7%) | 9 (5.7%) | 136 (22.6%) | <0.001 |
| *Unilateral/bilateral pulmonary infiltrates* | 17 (20.0%) | 62 (54.9%) | 29 (85.3%) | 108 (46.6%) | 87 (48.9%) | 212 (79.7%) | 146 (93.0%) | 445 (74.0%) | <0.001 |
| *Pleural effusion* | 3 (3.5%) | 11 (9.7%) | 7 (20.6%) | 21 (9.1%) | 5 (2.8%) | 10 (3.8%) | 5 (3.2%) | 20 (3.3%) | 0.01 |
| *Pulmonary edema* | 2 (1.3%) | 8 (6.6%) | 3 (7.9%) | 13 (4.1%) | 1 (0.4%) | 4 (1.4%) | 5 (3%) | 10 (1.4%) | 0.01 |
| *Other findings* | 9 (10.6%) | 11 (7.1%) | 2 (5.9%) | 22 (5.6%) | 9 (5.1%) | 10 (3.8%) | 4 (2.5%) | 23 (3.8%) | 0.01 |
| **Chest CT scan performed in** | 18 (11.4%) | 20 (17.7%) | 13 (38.2%) | 51 (16.1%) | 20 (7.7%) | 41 (13.9%) | 73 (43.2%) | 134 (18.5%) | <0.001 |
| **Chest CT scan findings** | | | | | | | | | |
| *Normal CT* | 8 (5.1%) | 3 (2.5%) | | 11 (3.5%) | 3 (1.1%) | 1 (0.3%) | | 4 (0.6%) | 0.29 |
| *Unilateral subpleural ground glass opacities* | | | 1 (2.6%) | 1 (0.3%) | 4 (1.5%) | 3 (1%) | 1 (0.6%) | 8 (1.1%) | <0.001 |
| *Bilateral subpleural ground glass opacities with or without consolidation* | 2 (1.3%) | 6 (5%) | 8 (21.1%) | 16 (5%) | 12 (4.6%) | 32 (10.8%) | 68 (40.2%) | 112 (15.4%) | <0.001 |
| *Unilateral infiltrates with consolidation* | 3 (1.9%) | | 2 (5.3%) | 5 (1.6%) | | 2 (0.7%) | | 2 (0.3%) | 0.03 |
| *Pulmonary thromboembolism (central or peripheral)* | 2 (1.3%) | 2 (1.7%) | | 4 (1.3%) | 2 (0.8%) | 2 (0.7%) | 15 (8.9%) | 19 (2.6%) | 0.25 |
| *Plural effusion* | 3 (1.9%) | 3 (2.5%) | 2 (5.3%) | 8 (2.5%) | 1 (0.4%) | 3 (1%) | 6 (3.6%) | 10 (1.4%) | 0.20 |
| *Other CT* | 4 (2.5%) | 9 (7.4%) | 6 (15.8%) | 19 (6%) | 3 (1.1%) | 5 (1.7%) | 3 (1.8%) | 11 (1.5%) | <0.001 |

*Peak value during admission.

**PCR positive (throat swab or endotracheal aspirate) for *Legionella pneumophila*, *Mycoplasma pneumoniae*, or *Chlamydia pneumoniae*.

resulted from the 60-day risk of death analysis. The main findings from the study are summarized in Table 4 and Fig 1. Excluding patients who received three vaccine doses did not meaningfully change the results of these analyses (results not shown).

Similar hazard ratios for severe hypoxia, 30- and 60-day mortality were obtained in the analyses of the 316 individually matched patients (Tabel 4 and Supplementary Tables A–C in S1 File). In the subanalysis where interaction between the variants and vaccination status was

**Table 3. Treatment, clinical outcomes and mortality of patients hospitalized with COVID-19 due to SARS-CoV-2 Omicron and Delta variants, according to disease severity (Copenhagen, 1 September 2021 to 11 February 2022; n = 1,043).**

| | | Omicron | | | | Delta | | | | |
| --- | --- | --- | --- | --- | --- | --- | --- | --- | --- | --- |
| | | Mild/ Moderate | Severe | Critical | All | Mild/ Moderate | Severe | Critical | All | *p*-value |
| **Total** | | 158 (49.8%) | 121 (38.2%) | 38 (12%) | 317 (100%) | 261 (36%) | 296 (40.8%) | 169 (23.3%) | 726 (100%) | |
| **Oxygen supplementation** | *No oxygen* | 148 (93.7%) | 5 (4.1%) | | 153 (48.3%) | 247 (94.6%) | 7 (2.4%) | | 254 (35%) | <0.001 |
| | *Less than 10 L/min* | 10 (6.3%) | 102 (84.3%) | 1 (2.6%) | 113 (35.6%) | 12 (4.6%) | 261 (88.2%) | 3 (1.8%) | 276 (38%) | |
| | *More than 10 L/min administered by either HFNC, NIV or CPAP* * | | 14 (11.6%) | 37 (97.4%) | 51 (16.1%) | 2 (0.8%) | 28 (9.5%) | 166 (98.2%) | 196 (27%) | |
| **Remdesivir** | *Yes* | 35 (22.2%) | 89 (73.6%) | 25 (65.8%) | 149 (47%) | 35 (13.4%) | 218 (73.6%) | 128 (75.7%) | 381 (52.5%) | 0.12 |
| **Dexamethasone** | *Yes* | 24 (15.2%) | 105 (86.8%) | 33 (86.8%) | 162 (51.1%) | 38 (14.6%) | 269 (90.9%) | 164 (97%) | 471 (64.9%) | <0.001 |
| **Antibiotics** | *Yes* | 46 (29.1%) | 80 (66.1%) | 35 (92.1%) | 161 (50.8%) | 93 (35.6%) | 162 (54.7%) | 148 (87.6%) | 403 (55.5%) | 0.10 |
| **IL-6 receptor blocker** | *Yes* | | 1 (0,8%) | 8 (21,1%) | 9 (2.8%) | | | 48 (28,4%) | 48 (6.6%) | 0.01 |
| **Anticoagulants** | *None* | 103 (65.2%) | 18 (14.9%) | 4 (10.5%) | 125 (39.4%) | 171 (65.5%) | 26 (8.8%) | 2 (0.8%) | 200 (27.5%) | <0.001 |
| | *Prophylactic dosage* | 40 (25.3%) | 81 (66.9%) | 20 (52.6%) | 141 (44.5%) | 72 (27.6%) | 242 (81.8%) | 144 (85.2%) | 458 (63.1%) | |
| | *Therapeutic dosage* | 5 (3.2%) | 7 (5.8%) | 10 (26.3%) | 22 (6.9%) | 9 (3.4%) | 12 (4.1%) | 15 (8.9%) | 36 (5%) | |
| | *other dosage* | 10 (6.3%) | 14 (11.6%) | 4 (10.5%) | 28 (8.8%) | 8 (3.1%) | 13 (4.4%) | 3 (1.8%) | 24 (3.3%) | |
| | *Missing* | | 1 (0,8%) | | 1 (0.3%) | 1 (0,4%) | 3 (1%) | 4 (2,4%) | 8 (1.1%) | |
| **Monoclonal antibodies** | *No* | 137 (86.7%) | 106 (87.6%) | 31 (81.6%) | 274 (86.4%) | 217 (83.1%) | 267 (90.2%) | 141 (83.4%) | 625 (86.1%) | 0.36 |
| | *Yes, pre-admission* | 3 (1,9%) | | 1 (2,6%) | 4 (1.3%) | 4 (1,5%) | | | 4 (0.6%) | |
| | *Yes, during admission* | 17 (10.8%) | 13 (10.7%) | 6 (15.8%) | 36 (11.4%) | 38 (14.6%) | 29 (9.8%) | 26 (15.4%) | 93 (12.8%) | |
| | *Missing* | 1 (0,6%) | 2 (1,7%) | | 3 (0.9%) | 2 (0,8%) | | 2 (1,2%) | 4 (0.6%) | |
| **MIS-A**** | *No* | 158 (100%) | 116 (95.9%) | 37 (97.4%) | 311 (98.1%) | 254 (97.3%) | 290 (98%) | 165 (97.6%) | 709 (97.7%) | - |
| | *Missing information* | | 5 (4,1%) | 1 (2,6%) | 6 (1.9%) | 7 (2,7%) | 6 (2%) | 4 (2,4%) | 17 (2.3%) | |
| **Admission to ICU***** | *Yes* | | 3 (2,5%) | 9 (23,7%) | 12 (3.8%) | 1 (0,4%) | 3 (1%) | 73 (43,2%) | 77 (10.6%) | <0.001 |
| | *Mechanical ventilation* | | | 6 (100%) | 6 (1.9%) | | | 45 (100%) | 45 (6.2%) | <0.01 |
| | *ICU length of stay (days)* | | 2 (1–3) | 12 (8–19) | 8 (3.5–13.8) | 1 (1–1) | 2.5 (2.2–2.8) | 9 (3.5–15.5) | 8.5 (3–14.8) | 0.60 |
| | *Length of mechanical ventilation (days)* | | | 10 (3–11.8) | 10 (3–11.8) | | | 10 (6–15) | 10 (6–15) | 0.15 |
| **Length of stay, median (IQR) in days** | | 2 (1–4) | 5 (3–7) | 8.5 (4.2–13) | 4 (1–6) | 2 (1–3) | 4 (3–7) | 13 (8–19) | 4 (2–9) | <0.001 |
| **Readmission within 30 days** | | 17 (10.8%) | 14 (11.6%) | 7 (18.4%) | 38 (12%) | 36 (13.8%) | 45 (15.2%) | 12 (7.1%) | 93 (12.8%) | 0.73 |
| **Death within 30 days of admission** | | 1 (0.6%) | 21 (17.4%) | 23 (60.5%) | 45 (14.2%) | 10 (3.8%) | 38 (12.8%) | 51 (30.2%) | 99 (13.6%) | 0.85 |

(*Continued*)

**Table 3.** (Continued)

|  |  | Omicron | | | | Delta | | | | |
|---|---|---|---|---|---|---|---|---|---|---|
|  |  | Mild/ Moderate | Severe | Critical | All | Mild/ Moderate | Severe | Critical | All | *p*-value |
| **Death within 60 days of admission** |  | 3 (1.9%) | 27 (22.3%) | 23 (60.5%) | 53 (16.7%) | 12 (4.6%) | 48 (16.2%) | 58 (34.3%) | 118 (16.3%) | 0.86 |

\* high-flow nasal cannula, non-invasive ventilation or continuous positive airway pressure

\*\*Some numbers are masked due to privacy guidelines.

\*\*\*multisystem inflammatory syndrome in adults

\*\*\*\*intensive care unit

included, among those who had received three vaccine doses, patients with Omicron had a lower risk of death at 30 and 60 days than patients with Delta (aHR, 0.42; 0.18–0.96 and 0.40; 0.19–0.87)). There was no statistically significant difference between the variants for those who received two doses (aHR, 0.88; 0.32–2.46 and 1.02; 0.4–2.61) or 0–1 dose (aHR, 0.72; 0.26–2.05 and 0.89; 0.32–2.44; Table 4). The interaction was only statistically significant for 60-day mortality (*p* = 0.07 for 30 days and *p* = 0.03 for 60 days).

## Discussion

In this cohort of adults hospitalized due to COVID-19, we found that the SARS-CoV-2 Omicron variant was associated with nearly 40% improved 30- and 60-day survival compared to patients hospitalized with the Delta variant. This was mainly driven by a decreased disease severity observed in Omicron patients vaccinated with three doses of an mRNA vaccine. However, there was no statistically significant difference in mortality risk between Omicron and Delta variants for unvaccinated patients or those who had received one or two vaccine doses. In addition, unadjusted mortality rates at 30 and 60 days were similar for both variants. Omicron patients were older and more frail than Delta patients. Our cohort included many older adults, had free access to health care and testing, and exhibited high COVID-19 vaccination coverage.

In a large population-based study from the UK [7], the risk of death occurring 0–28 days after SARS-CoV-2 infection was substantially lower for Omicron than for Delta, including for the unvaccinated. Our study addresses disease severity and mortality in hospitalized COVID-19 patients caused by the Omicron and Delta SARS-CoV-2 variants and has a follow-up of 60 days from hospitalization. A recent study of patients with solid and hematological malignancies found similar crude mortality rates in unvaccinated patients during the Omicron compared to the Alpha and Delta waves [27]. We found that risk of death for Omicron patients was similar for unvaccinated patients and those who received less than three doses; however, our risk estimates were around one, with wide CIs. Our findings suggest that high third vaccination-dose coverage may have prevented many deaths in our cohort.

We found that Omicron patients had a lower risk of severe hypoxemia, regardless of vaccination status. In addition, biological markers of inflammation and pulmonary involvement on CT scans were lower in the Omicron group. A study from France [8] reported similar results, although Omicron patients were younger, more often female, and better vaccinated than Delta patients. Finally, in agreement with other studies, we found that increasing age [27], frailty [28], and comorbidities [29] were strongly associated with mortality in hospitalized COVID-19 patients.

**Table 4. Univariate and multivariable hazard ratios for severe hypoxemia, 30-day all-cause mortality, and 60-day all-cause mortality, after COVID-19 hospital admission with Delta or Omicron SARS-CoV-2 variants in the total study population (*n* = 1043) and in the matched study population (*n* = 316).**

| | | Severe hypoxia^ | | 30-day mortality | | 60-day mortality | |
|---|---|---|---|---|---|---|---|
| | | Univariate HR | Multivariable HR | Univariate HR | Multivariable HR | Univariate HR | Multivariable HR |
| **Total population\*** | | | | | | | |
| **Omicron vs Delta** | | 0.55 (0.4–0.75) | 0.55 (0.38–0.78) | 1.04 (0.73–1.48) | 0.61 (0.39–0.95) | 1.03 (0.74–1.42) | 0.6 (0.4–0.91) |
| **CCI^^** | *0* | 1 (ref.) | 1 (ref.) | 1 (ref.) | 1 (ref.) | 1 (ref.) | 1 (ref.) |
| | *1–2* | 1.02 (0.77–1.35) | 1.12 (0.82–1.52) | 2.61 (1.7–4) | 1.51 (0.97–2.35) | 2.53 (1.72–3.73) | 1.45 (0.97–2.15) |
| | *3+* | 0.91 (0.64–1.29) | 1.15 (0.78–1.72) | 3.63 (2.29–5.76) | 2.09 (1.29–3.38) | 3.46 (2.27–5.27) | 1.88 (1.21–2.92) |
| **Frailty** | *No need for help on a daily basis* | 1 (ref.) | 1 (ref.) | 1 (ref.) | 1 (ref.) | 1 (ref.) | 1 (ref.) |
| | *Limited to full need for help on a daily basis* | 1.03 (0.71–1.5) | 1.31 (0.87–1.96) | 4.51 (3.2–6.37) | 3.14 (2.15–4.58) | 4.3 (3.12–5.93) | 3.03 (2.13–4.3) |
| **Age group** | *<60* | 1 (ref.) | 1 (ref.) | 1 (ref.) | 1 (ref.) | 1 (ref.) | 1 (ref.) |
| | *60–79* | 1.15 (0.85–1.56) | 1.4 (1–1.96) | 8.11 (3.72–17.69) | 6.99 (3.1–15.77) | 8.79 (4.25–18.2) | 7.22 (3.38–15.4) |
| | *80+* | 0.86 (0.61–1.21) | 1.16 (0.78–1.72) | 14.49 (6.68–31.46) | 11.49 (5.01–26.34) | 15.16 (7.34–31.28) | 11.26 (5.2–24.42) |
| **Sex** | *Male* | 1 (ref.) | 1 (ref.) | 1 (ref.) | 1 (ref.) | 1 (ref.) | 1 (ref.) |
| | *Female* | 0.63 (0.49–0.82) | 0.64 (0.49–0.83) | 0.77 (0.55–1.07) | 0.72 (0.51–1) | 0.75 (0.55–1.01) | 0.7 (0.52–0.96) |
| **Vaccination** | *Three doses* | 1 (ref.) | 1 (ref.) | 1 (ref.) | 1 (ref.) | 1 (ref.) | 1 (ref.) |
| | *Two doses* | 1.18 (0.81–1.71) | 0.84 (0.55–1.29) | 0.82 (0.55–1.23) | 0.81 (0.5–1.32) | 0.85 (0.6–1.22) | 0.82 (0.52–1.28) |
| | *None or one dose* | 1.97 (1.38–2.81) | 1.67 (1.09–2.57) | 0.67 (0.44–1.03) | 1.49 (0.9–2.45) | 0.55 (0.37–0.83) | 1.18 (0.74–1.89) |
| **Omicron vs. Delta by vaccination status ¤** | *Three doses* | 0.48 (0.25–0.89) | 0.48 (0.25–0.90) | 0.42 (0.22–0.8) | 0.31 (0.16–0.59) | 0.4 (0.23–0.72) | 0.29 (0.16–0.52) |
| | *Two doses* | 0.65 (0.33–1.25) | 0.69 (0.35–1.35) | 0.7 (0.33–1.48) | 0.86 (0.41–1.84) | 0.68 (0.35–1.32) | 0.85 (0.43–1.68) |
| | *None or one dose* | 0.61 (0.36–1.06) | 0.52 (0.30–0.90) | 2.24 (1.19–4.21) | 0.94 (0.49–1.81) | 2.44 (1.32–4.52) | 1.06 (0.56–2.00) |
| **Matched population\*\*** | | | | | | | |
| **Omicron vs Delta** | | 0.69 (0.43–1.12) | 0.62 (0.38–1.02) | 0.83 (0.49–1.4) | 0.58 (0.33–1.02) | 0.84 (0.51–1.38) | 0.63 (0.37–1.06) |
| **Frailty at admission** | *No need for help on a daily basis* | 1 (ref.) | 1 (ref.) | 1 (ref.) | 1 (ref.) | 1 (ref.) | 1 (ref.) |
| | *Limited to full need for help on a daily basis* | 1.65 (0.93–2.94) | 1.9 (1.05–3.4) | 4.02 (2.33–6.94) | 4.76 (2.68–8.44) | 3.36 (1.99–5.69) | 3.89 (2.24–6.75) |
| **Omicron vs. Delta by vaccination status ¤** | *Three doses* | 0.58 (0.23–1.45) | 0.54 (0.22–1.36) | 0.53 (0.23–1.19) | 0.42 (0.18–0.96) | 0.48 (0.23–1.04) | 0.4 (0.19–0.87) |
| | *Two doses* | 0.86 (0.37–2.04) | 0.78 (0.33–1.84) | 1.17 (0.42–3.22) | 0.88 (0.32–2.46) | 1.28 (0.5–3.24) | 1.02 (0.4–2.61) |
| | *None or one dose* | 0.66 (0.31–1.42) | 0.57 (0.26–1.25) | 1.15 (0.42–3.16) | 0.72 (0.26–2.05) | 1.30 (0.48–3.48) | 0.89 (0.32–2.44) |

*The analyses in the total population were adjusted for age (>60, 60–79, 80+), sex, frailty at admission, Charlson comorbidity index (CCI) and vaccination status (0,1,2,3 doses). ¤ HR shown are for model with interaction between variants and vaccination status. For the univariate analysis this only included variant, vaccination status, and their interaction, while the multivariable results are adjusted for age group, sex, CCI and frailty

**Individuals were matched in a 1:1 ratio on age (10-year intervals), sex, underlying comorbidities considered to be risk factors for developing severe or critical COVID-19 (0, 1, 2, >2), and vaccination status (0, 1, 2, 3 doses) at least 14 days before admission. The analyses in the matched population were adjusted for frailty at admission.
^Severe hypoxia is defined as a composite outcome consisting of the need for ≥ 10 L/min of oxygen supplementation, high-flow nasal cannula, non-invasive ventilation, continuous positive airway pressure, mechanical ventilation, or intensive care unit admission. ^^Charlson Comorbidity Index

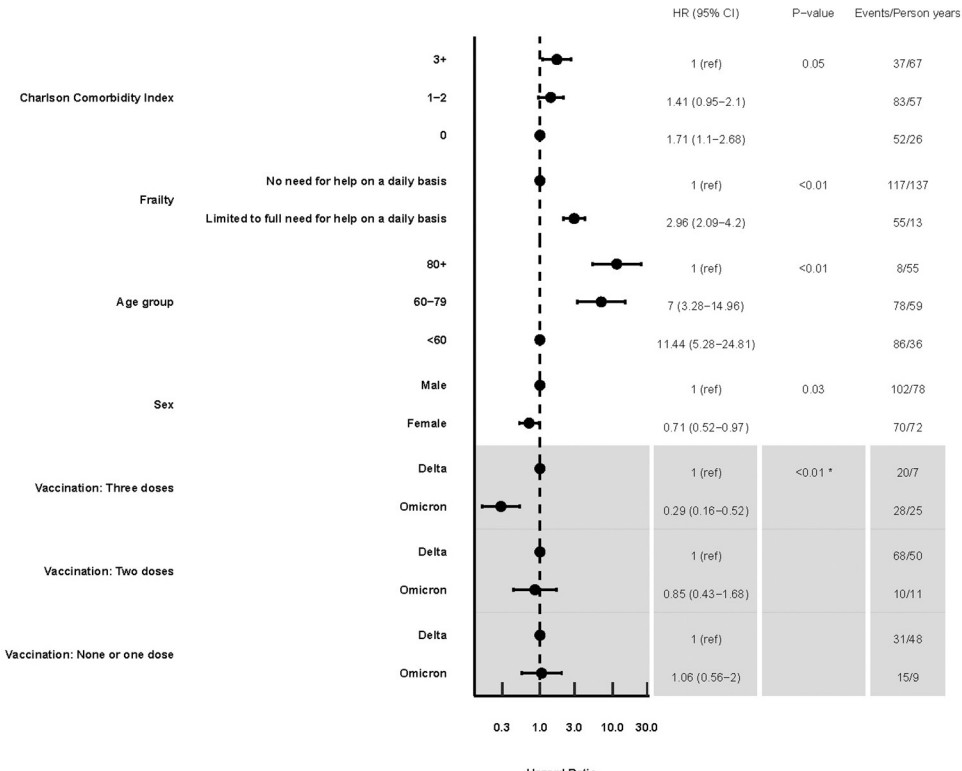

**Fig 1. Multivariate hazard ratios for 60-day mortality after COVID-19 hospital admission due to SARS-CoV-2 Delta or Omicron variants in the entire study population (*n* = 1,043).** *The grey area indicates the results where interaction between variants and vaccination status was included in the analysis. P-value for comparing this model with interaction to one without interaction.

Our study has limitations. First, high-risk patients (e.g., older, immunocompromised, or with severe chronic underlying diseases) were offered a third vaccine dose early during the fall, likely introducing some effect modification. Thus, patients hospitalized with the Delta variant who had received a third dose were more likely to be susceptible to poor clinical outcomes. We addressed this potential bias by evaluating interactions between vaccination status and SARS-CoV-2 variants and performing a subanalysis. We matched Omicron to Delta cases on risk factors, age, sex, and the number of vaccine doses. Even if we did not control for the time from last vaccination to admission, the time intervals between final vaccination and admission were similar for Omicron and Delta patients in the matched population. These analyses generated similar results for mortality but different results for severe hypoxemia. However, the CIs were wide, and this subanalysis likely lacked power. Second, relatively few individuals had 0–2 vaccine doses, resulting in wide CIs for these estimates.

The strengths of our study included the high-quality data, which were obtained from both medical records and linkage to national databases, the use of whole-genome sequencing and PCR for variant determination, a complete follow-up of the cohort, and the exclusion of incidental SARS-CoV-2 infected patients. Furthermore, our study population consisted of predominantly elderly multimorbid individuals, in whom increased virulence will have the greatest effect on outcomes. Therefore, our results are likely to be generalizable to other countries with similar populations and health care structures.

The finding that infection with the Omicron variant was associated with reduced mortality in hospitalized COVID-19 patients who had received three doses is reassuring for planning

vaccination strategies. Our results underline the importance of global public health efforts to achieve equity in access to COVID-19 vaccination in countries with low vaccination uptake.

## Supporting information

**S1 File.**
(DOCX)

## Acknowledgments

The full membership list of the **COVID-19 Omicron Delta study group:**

**Copenhagen University Hospital, North Zealand, Department of Pulmonary Medicine and Infectious Diseases:** MD, PhD Zitta Barrella Harboe*, MD, PhD Casper Roed, MD, PhD Jon G. Holler, MD Fahim Iqbal Khan, MD Aya Nihad Abdulrahman Abdulrahman, MD Stefan Lundby Mülverstedt, Betina Lindgaard-Jensen, MD Barbara Bonnesen Bertelsen, MD, PhD Christian Søborg, MD, PhD Thyge Lynghøj Nielsen, MD Line Vinum Hansen, MD, PhD Birgitte Lindegaard Madsen*, MD Andrea Browatzki, Mads Eiberg*, MD, PhD Peter Haahr Bernhard, MD Emilie Marie Juelstorp Pedersen*, MD, PhD Gertrud Baunbaek Egelund, MD Arnold Matovu Dungu*, MD Adin Sejdic*, MD, PhD Inger Hee Mabuza Mathiesen, MD, PhD Naja Z. Jespersen, MD Pelle Trier Petersen*, MD, PhD Lars Nielsen, MD, PhD Micha Phill Grønholm Jepsen, MD Thomas Ingemann Pedersen, MD, PhD Robert Eriksson, MD Hans Eric Sebastian Seitz-Rasmussen. **Copenhagen University Hospital, North Zealand, Department of Intensive Care:** MD, PhD Morten Bestle*, MD Henrik Andersen, MD Ulrik Skram, MD Mads Rømer Skøtt. **Copenhagen University Hospital, Herlev-Gentofte Hospital, Department of Pulmonary Medicine:** Sarah Altaraihi, MD Pradeesh Sivapalan, MD, PhD Jens-Ulrik Stæhr Jensen* **Copenhagen University Hospital, Amager-Hvidovre Hospital, Department of Clinical Microbiology:** MD Kristian Bagge, MD Kristina Melbardis Jørgensen, MD Maja Johanne Søndergaard Knudsen, MD Thomas Leineweber, MD, PhD Uffe Vest Schneider **Copenhagen University Hospital, Herlev-Gentofte Hospital, Department of Clinical Microbiology:** MD, PhD Magnus Glindvad Ahlström. **Copenhagen University Hospital, Herlev-Gentofte Hospital, Department of Internal Medicine, Section for Infectious Diseases:** MD Sofie Rytter, MD Nina le Dous, MD, PhD Pernille Ravn. **Copenhagen University Hospital, Bispebjerg-Frederiksberg Hospital, Department of Intensive Care:** MD Nanna Reiter. **Copenhagen University Hospital, Bispebjerg-Frederiksberg Hospital, Department of Respiratory Medicine and Infectious Diseases:** MD, PhD Daria Podlekareva, MD, PhD Andreas Knudsen, MD Stine Johnsen*. **Copenhagen University Hospital, Bispebjerg-Frederiksberg Hospital, Parker Institut:** MD, PhD Lars-Erik Kristensen*. **Copenhagen University Hospital, Amager-Hvidovre Hospital, Department of Infectious Diseases:** MD Cæcilie Leding, Bastian Bryan Hertz, MD, PhD Thomas Benfield*. **Copenhagen University Hospital, Rigshospitalet, Department of Infectious Diseases:** MD, PhD Ole Kirk*, MD, PhD Jon Gitz Holler. **Copenhagen University Hospital, Rigshospitalet, Department of Clinical Immunology**: MD, PhD Sisse Rye Ostrowski*. **Copenhagen University Hospital, Rigshospitalet, Department of Neurointensive Care:** MD Sigurdur Thor Sigurdsson **Copenhagen University Hospital, Rigshospitalet, Department of Intensive Care:** MD, PhD Anders Perner*. **Copenhagen University Hospital, Rigshospitalet, Department of Clinical Microbiology:** Nikolai Kirkby, PhD Martin Schou Pedersen. **PandemiX Center, Department of Science and Environment, Roskilde University:** PhD Maarten van Wijhe, PhD Lone Simonsen*. **Statens Serum Institut:** PhD Peter Michael Bager; MD, PhD Tyra Grove Krause; MD, PhD Marianne Voldstedlund; PhD Lasse Engbo Christiansen; PhD Marc Stegger; PhD

Arieh Cohen; PhD Jannik Fonager; MD, PhD Anders Fomsgaard; MD, PhD Rebecca Legarth; PhD Morten Rasmussen; PhD Sophie Gubbels; PhD Jan Wohlfahrt; MD, PhD Troels Lillebæk*. **Copenhagen University Hospital, North Zealand, Research Department:** MSc Caroline Klint Johannesen, PhD Maarten van Wijhe, MD, PhD Thea K Fischer** *These authors have a secondary affiliation to University of Copenhagen, Department of Clinical Medicine. **This author has a secondary affiliation to University of Copenhagen, Department of Public Health

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
