## [Decision Letter · Decision Letter 0]

29 Nov 2022

PONE-D-22-23300Clinical progression, disease severity, and mortality among adults hospitalized with COVID-19 caused by the Omicron and Delta SARS-CoV-2 variants: A population-based, matched cohort studyPLOS ONE

Dear Dr. Casper Roed,

Thank you for submitting your manuscript to PLOS ONE. After careful consideration, we feel that it has merit but does not fully meet PLOS ONE’s publication criteria as it currently stands. Therefore, we invite you to submit a revised version of the manuscript that addresses the points raised during the review process.

ACADEMIC EDITOR: the paper was revised by me and two reviewers. It requires minor revisions that need to be addressed. Please respond properly to reviewer comments.

We look forward to receiving your revised manuscript.

Kind regards,

Giuseppe Di Martino

Academic Editor

PLOS ONE

Journal Requirements:

2. "PLOS requires an ORCID iD for the corresponding author in Editorial Manager on papers submitted after December 6th, 2016. Please ensure that you have an ORCID iD and that it is validated in Editorial Manager. To do this, go to ‘Update my Information’ (in the upper left-hand corner of the main menu), and click on the Fetch/Validate link next to the ORCID field. This will take you to the ORCID site and allow you to create a new iD or authenticate a pre-existing iD in Editorial Manager. Please see the following video for instructions on linking an ORCID iD to your Editorial Manager account: " ext-link-type="uri" xlink:type="simple">https://www.youtube.com/watch?v=_xcclfuvtxQ"

Additional Editor Comments (if provided):

I revised the paper entitled "Clinical progression, disease severity, and mortality among adults hospitalized with COVID-19 caused by the Omicron and Delta SARS-CoV-2 variants: A population-based, matched cohort study". It was a cohort study aimed to evaluate differences in clinical outcomes between hospitalized patients with delta and omicron sars-cov-2 variant.

I want to congratulate with Authors for the excellent work. THe study was well conducted, the methodology was strong and deeply described.

I have only some minor observations:

- Did Authors considered the re-infection status? If yes, how it was handled in the analysis?

- was IperImmune Plasma therapy performed in Denmark?

- Among discussion, Author should better highlight that Omicron variant patients more frequently has three doses of vaccine, so it can strongly influence hospital outcomes;

- Did Authors considered to subanalyze by type of vaccination performed?

Reviewers' comments:

Reviewer's Responses to Questions

**Comments to the Author**

1. Is the manuscript technically sound, and do the data support the conclusions?

Reviewer #1: Yes

Reviewer #2: Yes

2. Has the statistical analysis been performed appropriately and rigorously? 

Reviewer #1: Yes

Reviewer #2: Yes

3. Have the authors made all data underlying the findings in their manuscript fully available?

Reviewer #1: Yes

Reviewer #2: Yes

4. Is the manuscript presented in an intelligible fashion and written in standard English?

Reviewer #1: Yes

Reviewer #2: Yes

5. Review Comments to the Author

Reviewer #1: Dear Authors:

I have read with interest the manuscript entitled "Clinical progression, disease severity, and mortality among adults hospitalized with COVID-19 caused by the Omicron and Delta SARS-CoV-2 variants: A population based, matched cohort study" for PlosOne to which I would like to make the following comments:

1. The manuscript the manuscript compares the virulence of the severe acute respiratory syndrome SARS-CoV-2 omicron variant with the delta variant in hospitalized adults with COVID-19 in the Capital Region of Copenhagen, Denmark.

2. The abstract summarizes well the characteristics and results of the study. Perhaps it could be emphasized in the conclusions section that the results have demonstrated the protective effect of vaccines by reducing the severity of the infection.

3. The introduction is correct, the characteristics of the Omicron variant and its differences with the Delta variant are exposed, since Omicron has a lower risk of hospitalization, intensive care unit (ICU) admission, and death compared to Delta. Through a cohort retrospective study, the differences between them are studied in hospitalized patients.

4. The methodology is correctly designed, the study design (retrospective, multicenter, matched cohort study) well explained, as well as the setting and data collection. The selection of patients, the variables, including the vaccination status, and the statistical analysis are correct. Ethical considerations are included.

5. The results are well expressed with the support of the 4 tables and figure 1. Table 4 and figure 1 should be highlighted as they make clear both the influence of vaccination (3 doses) and the risks of severe hypoxemia during admission and death at 30 and 60 days

6. The discussion is correct and highlights the results obtained, the limitations of the study are included. The conclusion: "the finding that infection with the Omicron variant was associated with reduced mortality in hospitalized COVID-19 patients who had received three doses is reassuring for planning vaccination strategies. Our results underline the importance of global public health efforts to achieve equity in access to COVID-19 vaccination in countries with low vaccination uptake", marks the direction of the path to follow emphatically.

7. The bibliographical references are adequate

Reviewer #2: This Danish multicentre observational study with a very high number of patients describes the difference in disease severity between the Delta and Omikron waves. The data are neatly presented and the manuscript is very well written, with the exceptions mentioned below. The observed effect of vaccination (with Omikron only one effect after three vaccinations) is, in my opinion, based on the effect observed in many other studies (including this Danish multicentre observational study with a very large number of patients). The data are neatly presented and the manuscript is very well written, with the exceptions mentioned below. The observed effect of vaccination (with Omikron only one effect after three vaccinations) is, in my opinion, based on the substantial effect of booster vaccination described in many other studies (including immunogenicity data). This should be emphasised more strongly.

1. Was the Sars-CoV-2 variant known in the patients? Some laboratories indicate the Sars-CoV-2 variant (at least Delta or Omicron) even without sequencing based on the PCR melting points. Please answer clearly in the text.

2. Due to the continuous transition from Delta to Omikron, a wash-out period of 2-4 weeks would have been necessary in my view. Why did the authors not do this? Please justify. This is a limitation.

3. This sentence from the abstract it's not clear to me, pls rephrase: „Omicron patients exhibited decreased aHR for 30-day mortality (aHR, 0.61; 0.39–0.95) and when given three vaccine doses (aHR, 0.31;0.16–0.59), but not two doses (aHR, 0.86; 0.41–1.84) or 0–1 dose (aHR, 0.94; 0.49–1.81).“ - To what does the vaccine status refer? Mortality within the Omicron patients?

4. The same applies to figure one. What does the grey area refer to? Is that the comparison of vaccine effectiveness (against mortality )within the Delta and within the Omikron patients? Was there no effect within the Delta group or was the effect within the Delta group set as reference?

5. Table 1: Why was the time since the last vaccination longer in the Delta group than in the Omikron group? At the time of Omikron, most patients should have received their third vaccination some time ago.

6. Table 2: The increased proportion of positive blood cultures in the Omikron group compared to Delta in critically ill patients is remarkable in my view. The proportion of bacterial co- infections in COVID-19 is increasing according to our experience. This is probably because the typical respiratory pathogens are returning and Sars-CoV-2 behaves the same as any other respiratory virus: bacterial co- infections are common and come in with increased severity. Was this increase significant? In any case, this aspect should be addressed in the discussion, as it has implications for future management.

7. Tab. 3: why did the critically ill patients with Omikron receive steroids less often than the critically ill patients under Delta?

8. Table 4: When comparing Omikron versus Delta by vaccination status, was the time elapsed since the last vaccination also taken into account?

6. PLOS authors have the option to publish the peer review history of their article (what does this mean?). If published, this will include your full peer review and any attached files.

Reviewer #1: No

Reviewer #2: **Yes: **Mathias Pletz

---

## [Author Response · Author response to Decision Letter 0]

17 Jan 2023

Response to Editors and Reviewers

We thank the reviewers and the editor for the insightful comments. We have responded to the individual points below and reviewed the manuscript accordingly. 

Additional Editor Comments (if provided):

I revised the paper entitled "Clinical progression, disease severity, and mortality among adults hospitalized with COVID-19 caused by the Omicron and Delta SARS-CoV-2 variants: A population-based, matched cohort study". It was a cohort study aimed to evaluate differences in clinical outcomes between hospitalized patients with delta and omicron sars-cov-2 variant. I want to congratulate with authors for the excellent work. The study was well conducted, the methodology was strong and deeply described. I have only some minor observations:

1. Did the authors consider the re-infection status? If yes, how it was handled in the analysis?

Response: Yes, we did consider re-infection status. Individuals with a prior positive RT-PCR SARS-CoV-2 test result were excluded from the study population. This exclusion is specified in the methods, subheading "Data sources and data collection", second paragraph: 

"We excluded individuals aged 18 years, those with a prior positive RT-PCR SARS-CoV-2 test result (…)"

Furthermore, this exclusion criterion is mentioned in the flow diagram (Supplementary Figure 1). 

2. Was hyperimmune plasma therapy performed in Denmark?

Response: No, hyperimmune plasma therapy was not recommended in Denmark at the time of the study. The following sentence has been added in the text, page 10, line 3: 

“Convalescent plasma was not recommended. Tixageivmab/cilgavivmab, bebtelovimab and nirmatrelvir/ritonavir were not yet available”

3. Among discussion, Author should better highlight that Omicron variant patients more frequently has three doses of vaccine, so it can strongly influence hospital outcomes

Response: Thank you for clarifying this. We have reworded the first paragraph of the discussion, page 14, so now it states: 

"In this cohort of adults hospitalized due to COVID-19, we found that the SARS-CoV-2 Omicron variant was associated with nearly 40% improved 30- and 60-day survival compared to patients hospitalized with the Delta variant. This was mainly driven by a decreased disease severity observed in Omicron patients vaccinated with three doses of an mRNA vaccine."

4. Did Authors considered to subanalyze by type of vaccination performed?

Response: A subanalysis by type of COVID-19 vaccine was not performed in our study because more than 90% of individuals in Denmark have received an mRNA vaccine. This was stated in the methods subheading "Setting";

"More than 90% of individuals received an mRNA vaccine17 (Comirnaty, BNT162b2 mRNA; BioNTech [Mainz, Germany]-Pfizer [New York, NY, USA] or Spikevax, mRNA-1273; Moderna, Cambridge, MA, USA)." 

Therefore we didn't consider performing a subanalysis by type of vaccination platform because a minority had received an adenovirus-based COVID-19 vaccine.

Reviewers' comments:

Reviewer #1: 

Dear Authors:

I have read with interest the manuscript entitled "Clinical progression, disease severity, and mortality among adults hospitalized with COVID-19 caused by the Omicron and Delta SARS-CoV-2 variants: A population based, matched cohort study" for PlosOne to which I would like to make the following comments:

1. The manuscript the manuscript compares the virulence of the severe acute respiratory syndrome SARS-CoV-2 omicron variant with the delta variant in hospitalized adults with COVID-19 in the Capital Region of Copenhagen, Denmark.

2. The abstract summarizes well the characteristics and results of the study. Perhaps it could be emphasized in the conclusions section that the results have demonstrated the protective effect of vaccines by reducing the severity of the infection. 

Response: Thank you for the comments. We have emphasized the importance of vaccination in the last sentence of the conclusion in the abstract, which states: 

"Among adults hospitalized with COVID-19, those with Omicron had less severe hypoxemia and nearly 40% higher 30- and 60-day survival, as compared with those with Delta, mainly driven by a larger proportion of Omicron patients vaccinated with three doses of an mRNA vaccine."

3. The introduction is correct, the characteristics of the Omicron variant and its differences with the Delta variant are exposed, since Omicron has a lower risk of hospitalization, intensive care unit (ICU) admission, and death compared to Delta. Through a cohort retrospective study, the differences between them are studied in hospitalized patients.

4. The methodology is correctly designed, the study design (retrospective, multicenter, matched cohort study) well explained, as well as the setting and data collection. The selection of patients, the variables, including the vaccination status, and the statistical analysis are correct. Ethical considerations are included.

5. The results are well expressed with the support of the 4 tables and figure 1. Table 4 and figure 1 should be highlighted as they make clear both the influence of vaccination (3 doses) and the risks of severe hypoxemia during admission and death at 30 and 60 days

Response: Thank you for this comment. In the results, it now states: 

"The main findings from the study are summarized in Table 4 and Figure 1." 

6. The discussion is correct and highlights the results obtained, the limitations of the study are included. The conclusion: "the finding that infection with the Omicron variant was associated with reduced mortality in hospitalized COVID-19 patients who had received three doses is reassuring for planning vaccination strategies. Our results underline the importance of global public health efforts to achieve equity in access to COVID-19 vaccination in countries with low vaccination uptake", marks the direction of the path to follow emphatically.

7. The bibliographical references are adequate

Reviewer #2: This Danish multicentre observational study with a very high number of patients describes the difference in disease severity between the Delta and Omikron waves. The data are neatly presented and the manuscript is very well written, with the exceptions mentioned below. The observed effect of vaccination (with Omikron only one effect after three vaccinations) is, in my opinion, based on the effect observed in many other studies (including this Danish multicentre observational study with a very large number of patients). The data are neatly presented and the manuscript is very well written, with the exceptions mentioned below. The observed effect of vaccination (with Omikron only one effect after three vaccinations) is, in my opinion, based on the substantial effect of booster vaccination described in many other studies (including immunogenicity data). This should be emphasised more strongly.

1. Was the Sars-CoV-2 variant known in the patients? Some laboratories indicate the Sars-CoV-2 variant (at least Delta or Omicron) even without sequencing based on the PCR melting points. Please answer clearly in the text.

Response: Thank you for this comment. One of the strengths of our study is that we only included patients with a known SARS-CoV-2 variant, determined at the national surveillance center through whole genome sequencing or a variant-specific RT-PCT. This is stated in the methods subheading "Data sources and data collection": 

Data from the National Patient Register20, the Danish Civil Register21, the Danish Vaccination Register22, and the national COVID-19 surveillance system at Statens Serum Institut (Copenhagen, Denmark)17,23 were used to identify all individuals aged ≥18 years who were hospitalized for 12 h within 14 days of a positive RT-PCR test for SARS-CoV-2 and from whom variant information was available. Surveillance and screening algorithms of SARS-CoV-2 variants in Denmark have been described elsewhere17,23. Variant determination was based on whole-genome sequencing or variant-specific RT-PCR performed at Statens Serum Institut or locally at the departments of clinical microbiology in the region. 

And additionally, under “Exposures”:

Exposures: Laboratory confirmed positive RT-PCR test results for Omicron or Delta SARS-CoV-2 variants.

2. Due to the continuous transition from Delta to Omikron, a wash-out period of 2-4 weeks would have been necessary in my view. Why did the authors not do this? Please justify. This is a limitation.

Response: Thank you for this comment related to the previous one. In contrast to most of the published studies, this study is not accounting for differences in outcomes between waves of different variants, we are comparing outcomes of patients with a known variant type. 

3. This sentence from the abstract it's not clear to me, pls rephrase: "Omicron patients exhibited decreased aHR for 30-day mortality (aHR, 0.61; 0.39–0.95) and when given three vaccine doses (aHR, 0.31;0.16–0.59), but not two doses (aHR, 0.86; 0.41–1.84) or 0–1 dose (aHR, 0.94; 0.49–1.81). "- To what does the vaccine status refer? Mortality within the Omicron patients?

Response: Thank you for this comment. Good point. We have attempted to clarify the abstract with the following modification: 

Omicron patients exhibited decreased aHR for 30-day mortality compared to Delta (aHR, 0.61; 039–0.95). Omicron patients who had received three vaccine doses had lower mortality compared to Delta patients who received three doses (aHR, 0.31;0.16–0.59), but not among those who received two or 0-1 doses (aHR, 0.86; 0.41–1.84 and 0.94; 0.49–1.81 respectively).

4. The same applies to figure one. What does the grey area refer to? Is that the comparison of vaccine effectiveness (against mortality) within the Delta and within the Omikron patients? Was there no effect within the Delta group or was the effect within the Delta group set as reference?

Response: Thank you for your comment. The grey area indicates the results where interaction between variants and vaccination status were included in the analysis. We have omitted the sentence "The grey area indicates interaction terms" in the figure title and instead added a footnote stating: The grey area indicates the results where interaction between variants and vaccination status were included in the analysis.

5. Table 1: Why was the time since the last vaccination longer in the Delta group than in the Omikron group? At the time of Omikron, most patients should have received their third vaccination some time ago.

Response: Thank you for your comment. The observation is correct regarding the timing of the third dose. The vaccination strategy for the third dose prioritized the elderly and immunocompromised individuals. Many industrialized countries followed a similar strategy, but the timing of the third dose differed widely. For example, in Denmark, many patients received their third vaccination during the omicron period. Therefore, we see a shorter time since the last vaccination in the omicron group than in the delta group. 

6. Table 2: The increased proportion of positive blood cultures in the Omikron group compared to Delta in critically ill patients is remarkable in my view. The proportion of bacterial co- infections in COVID-19 is increasing according to our experience. This is probably because the typical respiratory pathogens are returning and Sars-CoV-2 behaves the same as any other respiratory virus: bacterial co- infections are common and come in with increased severity. Was this increase significant? In any case, this aspect should be addressed in the discussion, as it has implications for future management.

Response: Thank you for your comment. This observation is very interesting. We decided not going into depth with this because we consider this would be the topic of a new manuscript with the rate of bacterial superinfections between variants.

7. Tab. 3: why did the critically ill patients with Omikron receive steroids less often than the critically ill patients under Delta?

Response: We reviewed the medical files for all patients: in the Omicron group, 4 out of 5 critically ill patients who did not receive dexamethasone were in a terminal state at time of admission. One of those patients received one dose of dexamethasone before death. The fifth patient was already on prednisolone 37.5 mg/dgl because of COPD exacerbation and the clinicians decided to continue with prednisolone instead of prescribing dexamethasone. 

8. Table 4: When comparing Omikron versus Delta by vaccination status, was the time elapsed since the last vaccination also taken into account?

Response: The time since the last vaccination was not considered in the analysis presented. However, we have done substantial checks on the influence of this timing. These checks were slightly restricted because the third dose was prioritized among the elderly during the omicron period (as mentioned earlier). This meant that ideal comparison groups could not be made, and many patients who had their last dose only several weeks prior to hospitalization were in the omicron group, while those with a longer time since the last vaccination were in the delta group. Nevertheless, when including time since vaccination as a categorical covariate at various levels, the results did not change meaningfully, and vaccination status and variant were still strong predictors for the outcome. 

Your sincerely 

Corresponding authors on behalf of all authors:

Zitta Barrella Harboe, Department of Pulmonary Medicine and Infectious Diseases Copenhagen University Hospital, North Zealand, Dyrehavevej 29, 3400 Hillerod: zitta.barrella.harboe@regionh.dk

Casper Roed, Department of Pulmonary Medicine and Infectious Diseases Copenhagen University Hospital, North Zealand, Dyrehavevej 29, 3400 Hillerod: casper.roed@regionh.dk

---

## [Decision Letter · Decision Letter 1]

23 Feb 2023

Clinical progression, disease severity, and mortality among adults hospitalized with COVID-19 caused by the Omicron and Delta SARS-CoV-2 variants: A population-based, matched cohort study

PONE-D-22-23300R1

Dear Dr. Casper Roed,

We’re pleased to inform you that your manuscript has been judged scientifically suitable for publication and will be formally accepted for publication once it meets all outstanding technical requirements.

Kind regards,

Giuseppe Di Martino

Academic Editor

PLOS ONE

Additional Editor Comments (optional):

Reviewers' comments:

Reviewer's Responses to Questions

**Comments to the Author**

1. If the authors have adequately addressed your comments raised in a previous round of review and you feel that this manuscript is now acceptable for publication, you may indicate that here to bypass the “Comments to the Author” section, enter your conflict of interest statement in the “Confidential to Editor” section, and submit your "Accept" recommendation.

Reviewer #1: All comments have been addressed

Reviewer #2: All comments have been addressed

2. Is the manuscript technically sound, and do the data support the conclusions?

Reviewer #1: Yes

Reviewer #2: Yes

3. Has the statistical analysis been performed appropriately and rigorously? 

Reviewer #1: Yes

Reviewer #2: Yes

4. Have the authors made all data underlying the findings in their manuscript fully available?

Reviewer #1: Yes

Reviewer #2: Yes

5. Is the manuscript presented in an intelligible fashion and written in standard English?

Reviewer #1: Yes

Reviewer #2: Yes

6. Review Comments to the Author

Reviewer #1: Dear Authors: Your manuscript is well written, with interesting results, and can be published in its present form

Reviewer #2: You have nicely addressed my issues. I encourage you to take a deeper look into the bacterial co-infections.

7. PLOS authors have the option to publish the peer review history of their article (what does this mean?). If published, this will include your full peer review and any attached files.

Reviewer #1: No

Reviewer #2: No

---

## [Editor Report · Acceptance letter]

28 Feb 2023

PONE-D-22-23300R1 

Clinical progression, disease severity, and mortality among adults hospitalized with COVID-19 caused by the Omicron and Delta SARS-CoV-2 variants: A population-based, matched cohort study 

Dear Dr. Roed:

I'm pleased to inform you that your manuscript has been deemed suitable for publication in PLOS ONE. Congratulations! Your manuscript is now with our production department. 

Kind regards, 

on behalf of

Dr. Giuseppe Di Martino 

Academic Editor

PLOS ONE